behaviour, cognition, theoretical biology

evolution of cooperation, reciprocity, Bayesian inference, Win-Stay-Lose-Shift, observational learning

**Author for correspondence:**
Seung Ki Baek
e-mail: seungki@pknu.ac.kr

# Win-Stay-Lose-Shift as a self-confirming equilibrium in the iterated Prisoner's Dilemma

Minjae Kim[1], Jung-Kyoo Choi[2] and Seung Ki Baek[1]

[1]Department of Physics, Pukyong National University, Busan 48513, Korea
[2]Department of Economics, Kyungpook National University, Daegu 41566, Korea

 SKB, 0000-0002-4143-1187

Evolutionary game theory assumes that players replicate a highly scored player's strategy through genetic inheritance. However, when learning occurs culturally, it is often difficult to recognize someone's strategy just by observing the behaviour. In this work, we consider players with memory-one stochastic strategies in the iterated Prisoner's Dilemma, with an assumption that they cannot directly access each other's strategy but only observe the actual moves for a certain number of rounds. Based on the observation, the observer has to infer the resident strategy in a Bayesian way and chooses his or her own strategy accordingly. By examining the best-response relations, we argue that players can escape from full defection into a cooperative equilibrium supported by Win-Stay-Lose-Shift in a self-confirming manner, provided that the cost of cooperation is low and the observational learning supplies sufficiently large uncertainty.

## 1. Introduction

Evolutionary game theorists often assume that behavioural traits can be genetically transmitted across generations [1]. Along this line, researchers have investigated the genetic basis of cooperative behaviour [2,3]. However, humans learn many culture-specific behavioural rules through observational learning [4], and this mechanism mediates 'cultural' transmission that has been proved to exist among a number of non-human animals as well [5,6]. The mirror neuron research suggests that the primate brain may even have a specialized circuit for imitating each other's behaviour, which facilitates social learning [7–9]. In comparison with the direct genetic transmission, the non-genetic inheritance through social learning can provide better adaptability by responding faster to environmental changes [10].

In contrast with genetic inheritance, however, observational learning may lead to imperfect mimicry if observation is not sufficiently informative or involved with a systematic bias. The notion of self-confirming equilibrium (SCE) has been proposed by incorporating such imperfectness of observation in learning [11]: when an SCE strategy is played, some of the possible information sets may not be reached, so the players do not have exact knowledge but only certain untested belief about what their co-players would do at those unreached sets. It is nevertheless sustained as an equilibrium in the sense that no player can expect a better payoff by unilaterally deviating from it once given such belief, and that the beliefs do not conflict with observed moves. Dynamics of learning based on a limited set of information has been investigated in the context of the coordination game [12,13], in which the opponent's observed decision is assumed to be his or her strategy. However, the subtlety of cultural transmission manifests itself clearly when a strategy is regarded as a decision rule, hidden from the observer, rather than the decision itself.

In this work, we investigate the iterated Prisoner's Dilemma (PD) game among players with memory-one strategies, who infer the resident strategy

from observation and optimizes their own strategies against it. By memory-one, we mean that a player refers to the previous round to choose a move between cooperation and defection [14]. If we restrict ourselves to memory-one strategies, it is already well known in evolutionary game theory that 'Win-Stay-Lose-Shift (WSLS)' [15–17] can appear through mutation and take over the population from defectors if the cost of cooperation is low [14]. Compared with such an evolutionary approach, we will impose 'less bounded' rationality in that our players are assumed to be capable of computing the best response to a given strategy within the memory-one pure-strategy space. We will identify the best-response dynamics in this space and examine how the dynamics should be modified when observational learning introduces uncertainty in Bayesian inference about strategies. If every player exactly replicated each other's strategy, full defection would be a Nash equilibrium (NE) for any cost of cooperation. Under uncertainty in observation, however, our finding is that defection is not always an SCE so that the population can move to a cooperative equilibrium supported by WSLS, which is both an SCE and an NE and can thus be called a SCENE.

## 2. Method and result

### (a) Best-response relations without observational uncertainty

Let us define the one-shot PD game in the following form:

$$\begin{pmatrix} & C & D \\ \hline C & 1-c & -c \\ D & 1 & 0 \end{pmatrix}, \tag{2.1}$$

where we abbreviate cooperation and defection as $C$ and $D$, respectively, and $c$ is the cost of cooperation assumed to be $0 < c < 1$. In this work, the game of equation (2.1) will be repeated indefinitely. Furthermore, the environment is noisy: Even if a player intends to cooperate, it can be misimplemented as defection, or vice versa, with probability $\epsilon$. In the analysis below, we will take $\epsilon$ as an arbitrarily small positive number.

We will restrict ourselves to the space of memory-one ($M_1$) pure strategies. By a $M_1$ pure strategy, we mean that it chooses a move between $C$ and $D$ as a function of the two players' moves in the previous round. We thus describe such a strategy as $[p_{CC}, p_{CD}, p_{DC}, p_{DD}]$, where $p_{XY} = 1$ means that $C$ is prescribed when the players did $X$ and $Y$, respectively, in the previous round, and $p_{XY} = 0$ if $D$ is prescribed in the same situation. Note that the initial move in the first round is irrelevant to the long-term average payoff in the presence of error so that it has been discarded in the description of a strategy. The set of $M_1$ pure strategies, denoted by $\Delta$, contains 16 elements from $\mathbf{d}_0 \equiv [0, 0, 0, 0]$ to $\mathbf{d}_{15} \equiv [1, 1, 1, 1]$.

Let us assume that a player, say, Alice, takes a $M_1$ pure strategy $\mathbf{d}_\alpha$ as her strategy. The noisy environment effectively modifies her behaviour to

$$\mathbf{s}_A^\epsilon \equiv (1 - \epsilon)\mathbf{d}_\alpha + \epsilon(\mathbf{1} - \mathbf{d}_\alpha) \tag{2.2}$$

as if she were playing a mixed strategy, where $\mathbf{1} \equiv [1, 1, 1, 1]$. Likewise, Alice's co-player Bob chooses $\mathbf{d}_\beta$, and his effective behaviour is described by

$$\mathbf{s}_B^\epsilon \equiv (1 - \epsilon)\mathbf{d}_\beta + \epsilon(\mathbf{1} - \mathbf{d}_\beta). \tag{2.3}$$

The repeated interaction between Alice and Bob is Markovian, and it is straightforward to obtain the stationary probability distribution

$$\mathbf{v}(\mathbf{d}_\alpha, \mathbf{d}_\beta, \epsilon) = (v_{CC}, v_{CD}, v_{DC}, v_{DD}), \tag{2.4}$$

where $v_{XY}$ means the long-term average probability to observe Alice and Bob choosing $X$ and $Y$, respectively [18–20] (see appendix A for more details). The presence of $\epsilon > 0$ guarantees the uniqueness of $\mathbf{v}$. Alice's long-term average payoff against Bob is then calculated as

$$\Pi(\mathbf{d}_\alpha, \mathbf{d}_\beta, \epsilon) = \mathbf{v} \cdot \mathbf{P}, \tag{2.5}$$

where $\mathbf{P} \equiv (1 - c, -c, 1, 0)$ is a payoff vector corresponding to equation (2.1). As long as Alice can exactly identify Bob's strategy $\mathbf{d}_\beta$ with no observational uncertainty, she can find the best response to Bob within the set of $M_1$ pure strategies by applying every $\mathbf{d}_\alpha \in \Delta$ to equation (2.5).

In table 1, we list the best response to each strategy in $\Delta$ in the limit of small $\epsilon$ (see also figure 1 for its graphical representation). In most cases, the best-response dynamics ends up with $\mathbf{d}_0 = [0, 0, 0, 0]$, which is the best response to itself and often called Always-Defect (AllD). For example, if we start with Tit-for-Tat (TFT), represented as $\mathbf{d}_{10} = [1, 0, 1, 0]$, table 1 shows that the best response to TFT within $\Delta$ is Always-Cooperate (AllC), represented as $\mathbf{d}_{15} = [1, 1, 1, 1]$, to which AllD is the best response for obvious reasons.

However, two exceptions exist: The first one is $\mathbf{d}_8 = [1, 0, 0, 0]$, which we may call $M_1$ Grim Trigger ($GT_1$). If $c > 1/3$, this strategy is the best response to itself, and it is an inefficient equilibrium giving each player an average payoff of $O(\epsilon)$. The other exception is WSLS, represented by $\mathbf{d}_9 = [1, 0, 0, 1]$, which is the best response to itself when $c \leq 1/2$. It is an efficient NE, at which each player earns $1 - c + O(\epsilon)$ per round on average.

### (b) Observational learning

Now, let us imagine a monomorphic population of players who have adopted a strategy $\mathbf{d}_\gamma$ in common. The population is in equilibrium in the sense that a large ensemble of their states $XY \in \{CC, CD, DC, DD\}$ can represent the stationary probability distribution $\mathbf{v}(\mathbf{d}_\gamma, \mathbf{d}_\gamma, \epsilon)$. We have an observer, say, Alice, with a potential strategy $\mathbf{d}_\alpha$. As we learn social norms in childhood, it is assumed that Alice does not yet participate in the game but has a learning period to observe $M$ ($\gg 1$) pairs of players, all of whom have used the resident strategy $\mathbf{d}_\gamma$. How their mind works is a black box to her: Just by observing their states $XY$ and subsequent moves, Alice has to form belief about $\mathbf{d}_\gamma$, based on which she chooses her own strategy $\mathbf{d}_\alpha$ to maximize the expected payoff. If Alice's optimal strategy turns out to be identical to the resident strategy $\mathbf{d}_\gamma$, it constitutes an SCE.

To see how Alice can specify $\mathbf{d}_\gamma \in \Delta$ from observation, let us consider an example that the observed probability distribution over states $XY$ is best described as $\mathbf{v} \approx (0, 1/4, 1/4, 1/2)$. If Alice has computed $\mathbf{v}$ for every strategy in $\Delta$ as listed in table 2, the observation suggests that the resident strategy is unlikely to be TFT ($\mathbf{d}_{10} = [1, 0, 1, 0]$) because the corresponding stationary distribution would be $\mathbf{v} = (1/4, 1/4, 1/4, 1/4)$. She finds that $\mathbf{d}_\gamma$ can be either $\mathbf{d}_2 = [0, 0, 1, 0]$ or $\mathbf{d}_4 = [0, 1, 0, 0]$. To distinguish between them, she has to check how people react to $CD$ or $DC$. According to table 2, these states will be observed frequently because $v_{CD} = v_{DC} =$

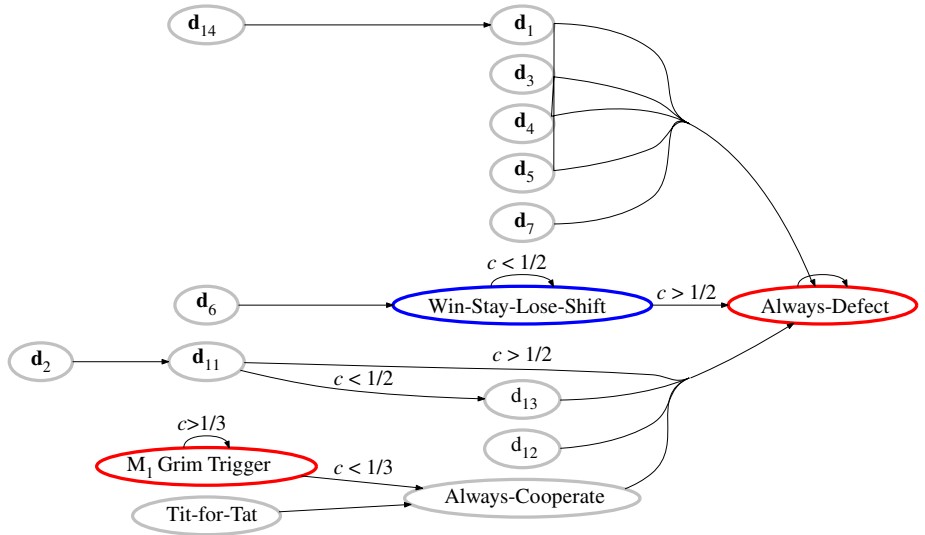

**Figure 1.** Graphical representation of best-response relations in table 1. If $\mathbf{d}_\mu$ is the best response to $\mathbf{d}_\nu$, we represent it as an arrow from $\mathbf{d}_\nu$ to $\mathbf{d}_\mu$. The blue node (Win-Stay-Lose-Shift) means an efficient NE with $1 - v_{CC} \sim O(\epsilon)$, whereas the red nodes (Always-Defect and $M_1$ Grim Trigger) mean inefficient ones with $v_{CC} \lesssim O(\epsilon)$ as shown in table 2. (Online version in colour.)

**Table 1.** Best response among $M_1$ pure strategies. Against each strategy in the first column, we obtain the best response (the second column), and the resulting average payoff (equation (2.5)) earned by the best response is given as a power series of $\epsilon$ in the third column. In the second column, we have placed a dagger next to a strategy when it is the best response to itself.

| opponent strategy | best response | payoff of the best response to the opponent strategy | Misc. |
|---|---|---|---|
| $\mathbf{d}_0$ | $\mathbf{d}_0^\dagger$ | $(1-c)\epsilon$ | AllD |
| $\mathbf{d}_1$ | $\mathbf{d}_0$ | $1/2 - (1/4+c)\epsilon + O(\epsilon^2)$ | |
| $\mathbf{d}_2$ | $\mathbf{d}_{11}$ | $(1-c)/2 - (1+c)\epsilon/2 + O(\epsilon^2)$ | |
| $\mathbf{d}_3$ | $\mathbf{d}_0$ | $1/2 - ce + O(\epsilon^3)$ | |
| $\mathbf{d}_4$ | $\mathbf{d}_0$ | $1/3 + (2/9-c)\epsilon + O(\epsilon^2)$ | |
| $\mathbf{d}_5$ | $\mathbf{d}_0$ | $1 - (2+c)\epsilon + O(\epsilon^2)$ | |
| $\mathbf{d}_6$ | $\mathbf{d}_9$ | $1 - 3(1+c)\epsilon + O(\epsilon^2)$ | |
| $\mathbf{d}_7$ | $\mathbf{d}_0$ | $1 - (2+c)\epsilon + 4\epsilon^2 + O(\epsilon^3)$ | |
| $\mathbf{d}_8$ | $\begin{cases} \mathbf{d}_8^\dagger, & c > 1/3 \\ \mathbf{d}_{15}, & c < 1/3 \end{cases}$ | $\begin{cases} 3(1-c)\epsilon/2 + O(\epsilon^2) \\ 1/3 - c + O(\epsilon) \end{cases}$ | $GT_1$ |
| $\mathbf{d}_9$ | $\begin{cases} \mathbf{d}_0, & c > 1/2 \\ \mathbf{d}_9^\dagger, & c < 1/2 \end{cases}$ | $\begin{cases} 1/2 + O(\epsilon) \\ 1 - c + O(\epsilon) \end{cases}$ | WSLS |
| $\mathbf{d}_{10}$ | $\mathbf{d}_{15}$ | $(1-c) - (2-c)\epsilon + O(\epsilon^2)$ | TFT |
| $\mathbf{d}_{11}$ | $\begin{cases} \mathbf{d}_0, & c > 1/2 \\ \mathbf{d}_{13}, & c < 1/2 \end{cases}$ | $\begin{cases} 1/2 + (1/4-c)\epsilon + O(\epsilon^2) \\ (1-c) - (2-c)\epsilon + O(\epsilon^2) \end{cases}$ | |
| $\mathbf{d}_{12}$ | $\mathbf{d}_0$ | $1/2 + O(\epsilon)$ | |
| $\mathbf{d}_{13}$ | $\mathbf{d}_0$ | $1 - (1+c)\epsilon + O(\epsilon^2)$ | |
| $\mathbf{d}_{14}$ | $\mathbf{d}_1$ | $1 - 2(1+c)\epsilon + O(\epsilon^2)$ | |
| $\mathbf{d}_{15}$ | $\mathbf{d}_0$ | $1 - (1+c)\epsilon + O(\epsilon^3)$ | AllC |

1/4. Thus, in this example, Alice succeeds in identifying $\mathbf{d}_\gamma$ as long as $M \gg 1$. Eight strategies have this property, constituting Category I in $\Delta$ (table 2). As another example, if $\mathbf{v} \approx (1/2, 0, 0, 1/2)$, Alice sees that $\mathbf{d}_\gamma$ must be either $\mathbf{d}_1 = [0, 0, 0, 1]$ or $\mathbf{d}_7 = [0, 1, 1, 1]$. To resolve the uncertainty, she has to further check how people react to *CD* or *DC*, but she may actually save this effort because the best response turns out

to be $\mathbf{d}_0$ in either case (table 1). This is the case of Category II in $\Delta$ (table 2).

In general, the first important piece of information to infer $\mathbf{d}_\gamma$ is the stationary distribution $\mathbf{v}$ because it heavily depends on $\mathbf{d}_\gamma$ (table 2). However, the information of $\mathbf{v}$ may be insufficient to single out the answer: Suppose that $\mathbf{v}$ gives multiple candidate strategies which prescribe different moves at a

**Table 2.** Stationary probability distribution $\mathbf{v}(\mathbf{d}_\gamma, \mathbf{d}_\gamma, \epsilon)$, where we have retained only the leading-order term in the $\epsilon$-expansion for each $v_{XY}$. When we describe a strategy in binary, the boldface digits are the ones that are frequently observed with $v_{XY} \sim O(1)$ and thus readily identifiable as long as $M \gg 1$. In this table, the eight strategies in Category I have three or four such digits, so if the population is using one of these strategies, Alice can tell which one is being played after $M \ (\gg 1)$ observations. As for Category II, the member strategies $\mathbf{d}_1$ and $\mathbf{d}_7$ would be indistinguishable if $M \ll \epsilon^{-1}$ because they differ at their non-boldface digits. Still, Alice can find the best response $\mathbf{d}_0$ which is common to both of them (table 1). In Category III, each member strategy has just one boldface digit, so the strategies as well as the best responses can be identified only if $M \gg \epsilon^{-1}$.

| category | strategy | $v_{CC}$ | $v_{CD}$ | $v_{DC}$ | $v_{DD}$ |
|---|---|---|---|---|---|
| I | $\mathbf{d}_3 = [\mathbf{0}, \mathbf{0}, \mathbf{1}, \mathbf{1}]$ | $\frac{1}{4}$ | $\frac{1}{4}$ | $\frac{1}{4}$ | $\frac{1}{4}$ |
| | $\mathbf{d}_5 = [\mathbf{0}, \mathbf{1}, \mathbf{0}, \mathbf{1}]$ | | | | |
| | $\mathbf{d}_{10} = [\mathbf{1}, \mathbf{0}, \mathbf{1}, \mathbf{0}]$ | | | | |
| | $\mathbf{d}_{12} = [\mathbf{1}, \mathbf{1}, \mathbf{0}, \mathbf{0}]$ | | | | |
| | $\mathbf{d}_2 = [\mathbf{0}, \mathbf{0}, \mathbf{1}, 0]$ | $\frac{1}{2}\epsilon$ | $\frac{1}{4}$ | $\frac{1}{4}$ | $\frac{1}{2}$ |
| | $\mathbf{d}_4 = [\mathbf{0}, \mathbf{1}, \mathbf{0}, 0]$ | | | | |
| | $\mathbf{d}_{11} = [\mathbf{1}, \mathbf{0}, \mathbf{1}, \mathbf{1}]$ | $\frac{1}{2}$ | $\frac{1}{4}$ | $\frac{1}{4}$ | $\frac{1}{2}\epsilon$ |
| | $\mathbf{d}_{13} = [\mathbf{1}, \mathbf{1}, \mathbf{0}, \mathbf{1}]$ | | | | |
| II | $\mathbf{d}_1 = [\mathbf{0}, \mathbf{0}, 0, \mathbf{1}]$ | $\frac{1}{2}$ | $\epsilon$ | $\epsilon$ | $\frac{1}{2}$ |
| | $\mathbf{d}_7 = [\mathbf{0}, 1, 1, \mathbf{1}]$ | | | | |
| III | $\mathbf{d}_0 = [\mathbf{0}, 0, 0, 0]$ | $\epsilon^2$ | $\epsilon$ | $\epsilon$ | $1$ |
| | $\mathbf{d}_6 = [0, 1, 1, \mathbf{0}]$ | $2\epsilon$ | $\epsilon$ | $\epsilon$ | $1$ |
| | $\mathbf{d}_8 = [\mathbf{1}, 0, 0, 0]$ | $\frac{1}{2}\epsilon$ | $\epsilon$ | $\epsilon$ | $1$ |
| | $\mathbf{d}_9 = [\mathbf{1}, 0, 0, 1]$ | $1$ | $\epsilon$ | $\epsilon$ | $2\epsilon$ |
| | $\mathbf{d}_{14} = [\mathbf{1}, 1, 1, 0]$ | $1$ | $\epsilon$ | $\epsilon$ | $\frac{1}{2}\epsilon$ |
| | $\mathbf{d}_{15} = [\mathbf{1}, 1, 1, 1]$ | $1$ | $\epsilon$ | $\epsilon$ | $\epsilon^2$ |

certain state $XY$ and thus have different best responses. Alice then needs to observe what players actually choose at $XY$, and such observations should be performed sufficiently many times, i.e. $M v_{XY} \gg 1$, for the sake of statistical power. If we check every $\mathbf{d}_\gamma \in \Delta$ one by one in this way, we see that the best response to the resident strategy can readily be identified as long as $M \gg \epsilon^{-1}$, in which case the result of observational learning would be the same as that of exact identification of strategies.

If $M \ll \epsilon^{-1}$, on the other hand, Alice cannot fully resolve such uncertainty through observation. Still, note that $M$ should be taken as far greater than $O(1)$ for statistical inference to be meaningful. Furthermore, $\epsilon$ has been introduced as a regularization parameter whose exact magnitude is irrelevant, so we look at the behaviour in the limit of small $\epsilon$. When $1 \ll M \ll \epsilon^{-1}$, uncertainty in the best response remains only when $\mathbf{v} \approx (0, 0, 0, 1)$ or $(1, 0, 0, 0)$, both of which are characteristic of Category III in table 2. In the former case, $\mathbf{d}_0$, $\mathbf{d}_6$ and $\mathbf{d}_8$ are the candidate strategies for $\mathbf{d}_\gamma$, whereas in the latter case, the candidates are $\mathbf{d}_9$, $\mathbf{d}_{14}$ and $\mathbf{d}_{15}$. From the Bayesian perspective, it is reasonable to assign equal probability to each of the candidate strategies. However, if $M\epsilon \ll 1$, the number of observations cannot be enough to update this prior probability (see appendix B for a detailed discussion). Therefore, when $\mathbf{v} \approx (0, 0, 0, 1)$, yielding $\mathbf{d}_\gamma = \mathbf{d}_0$ or $\mathbf{d}_6$ or $\mathbf{d}_8$, Alice tries to maximize the

expected payoff

$$\overline{\Pi}_\alpha = \frac{\Pi(\mathbf{d}_\alpha, \mathbf{d}_0, \epsilon) + \Pi(\mathbf{d}_\alpha, \mathbf{d}_6, \epsilon) + \Pi(\mathbf{d}_\alpha, \mathbf{d}_8, \epsilon)}{3}, \tag{2.6}$$

and the calculation shows that it can be achieved by playing

$$\begin{cases} \mathbf{d}_8, & \text{if } c > \frac{16}{33} \\ \mathbf{d}_9, & \text{if } c < \frac{16}{33} \end{cases} \tag{2.7}$$

in the limit of $\epsilon \to 0$. Likewise, when $\mathbf{v} \approx (1, 0, 0, 0)$, yielding $\mathbf{d}_\gamma = \mathbf{d}_9$ or $\mathbf{d}_{14}$ or $\mathbf{d}_{15}$, Alice tries to maximize her expected payoff from the three possibilities, which is achieved when she plays

$$\begin{cases} \mathbf{d}_1, & \text{if } c > \frac{2}{9} \\ \mathbf{d}_9, & \text{if } c < \frac{2}{9} \end{cases} \tag{2.8}$$

as $\epsilon \to 0$. Now, AllD ceases to be the best-looking response to itself (figure 2): The expected payoff against AllD will be higher when WSLS is played, if $c < 16/33$. On the other hand, if we consider a WSLS population with $c < 2/9$, its cooperative equilibrium is protected from invasion of defectors because Alice under observational uncertainty will keep choosing WSLS, which is truly the best response to itself.

The above analysis concerns the uniform prior among three candidate strategies in each case. Let $f_i$ denote the fraction of $\mathbf{d}_i$. For an observer who almost always sees defection from the population, the prior in equation (2.6) can be written as $(f_0, f_6, f_8) = (1/3, 1/3, 1/3)$. For a general prior $(f_0, f_6, f_8)$ with $0 < f_i < 1$ and $f_8 = 1 - f_0 - f_6$, the condition for WSLS to give the highest expected payoff is summarized as the intersection of the following two inequalities (figure 3a):

$$f_6 > \frac{1}{3}f_8 - \left(\frac{5c}{4 + 3c}\right) \tag{2.9}$$

and

$$f_6 > \left(\frac{3c}{2 + 3c}\right) - \frac{3}{5}\left(\frac{2 - c}{2 + 3c}\right)f_8. \tag{2.10}$$

The above inequalities are written for $f_6$ because it is $\mathbf{d}_6$ that has WSLS as the best response (table 1). If $c > 1/3$, the former inequality becomes trivial because of the positivity of $f_6$. Note that WSLS still gives the highest expected payoff for a significant part of the simplex even when the cost of cooperation is as high as $c = 0.9$ (figure 3b).

Similarly, we can check what an observer would conclude after observing nearly cooperation only, although it is of less importance compared with the above case of a defecting population (figure 2). For a general prior represented by $(f_9, f_{14}, f_{15})$, where $f_{14} = 1 - f_9 - f_{15}$, WSLS gives the highest expected payoff when

$$f_9 > \left(\frac{c}{1 - c}\right)\left(1 + \frac{f_{15}}{2}\right), \tag{2.11}$$

as can be seen in figure 3c. This inequality can be satisfied only if $c \leq 1/2$: Otherwise, it is better to be a defector by playing $\mathbf{d}_0$ or $\mathbf{d}_1$ (figure 3d).

## 3. Summary and discussion

In summary, we have investigated the iterated PD game in terms of best-response relations and checked how it is modified by observational learning. Thereby we have addressed a question about how cooperation is affected by cultural

*Proc. R. Soc. B* **288**: 20211021

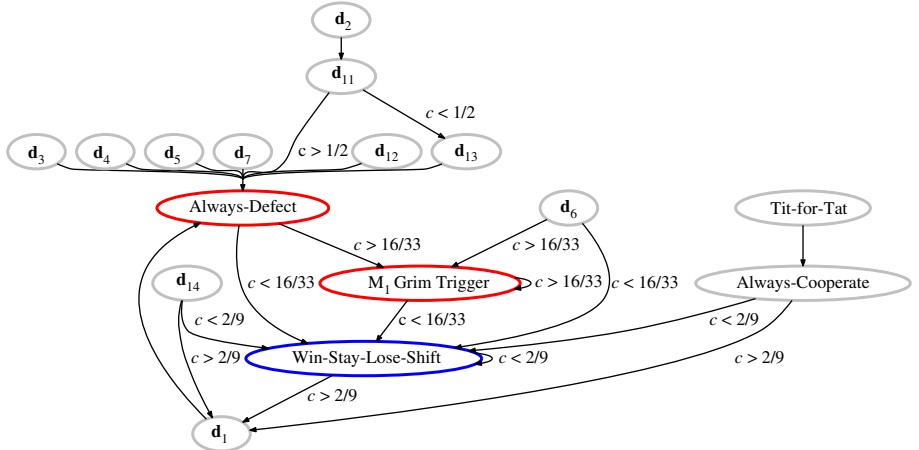

**Figure 2.** Best-looking responses to maximize the expected payoff under uncertainty in observation, when $1 \ll M \ll \epsilon^{-1}$. Compared with figure 1, the first difference is that Alice uses equation (2.7) against $\mathbf{d}_0$, $\mathbf{d}_6$ and $\mathbf{d}_8$. In addition, she will use equation (2.8) against $\mathbf{d}_9$, $\mathbf{d}_{14}$ and $\mathbf{d}_{15}$. (Online version in colour.)

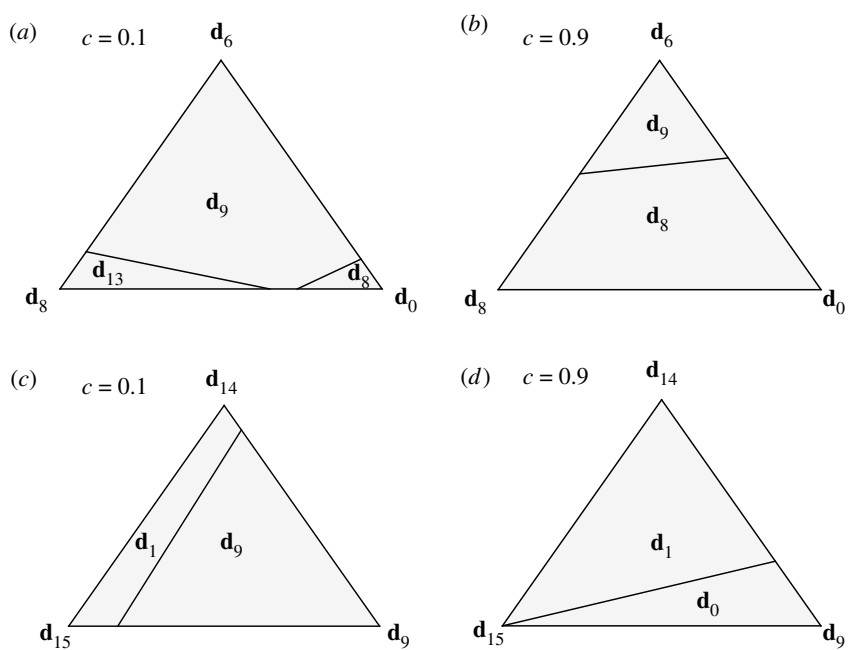

**Figure 3.** Effect of the prior on the observer's choice. A point in the triangle represents three fractions, which sum up to one, and its distance to an edge is proportional to the fraction of the strategy at the opposite vertex [21]. (*a*) When the observer sees nearly defection only, the prior takes the form of ($f_0$, $f_6$, $f_8$), for which we can find the strategy that gives the best expected payoff as written in each region. When $c$ is low, $\mathbf{d}_9$ (WSLS) gives the highest expected payoff for most of the prior. (*b*) Even when the cost increases to $c = 0.9$, the observer should choose WSLS if the prior contains a sufficiently high fraction of $\mathbf{d}_6$. (*c*) If the observer sees cooperation almost all the time, the prior can be expressed as ($f_9$, $f_{14}$, $f_{15}$). If $c$ is low, WSLS can be the observer's choice when $f_9$ is high enough. (*d*) The region of WSLS disappears as $c$ exceeds 1/2, and the only possible choice is between $\mathbf{d}_1$ and $\mathbf{d}_0$ (AllD).

transmission, which may be systematically involved with observational uncertainty. The notion of SCE takes this systematic uncertainty into account, and its intersection with NE can be an equilibrium refinement. It is worth pointing out the following: If everyone plays a certain strategy $\mathbf{d}_i$ with belief that everyone else does the same, the whole situation is self-consistent in the sense that observation will always confirm the belief, which in turn agrees with the actual behaviour. The importance of SCENE becomes clear when someone happens to play a different strategy or begins to doubt the belief: If $\mathbf{d}_i$ is not an NE, the player will benefit from the deviant behaviour and reinforce it. If $\mathbf{d}_i$ is not an SCE, the player may fail to dispel the doubt, which will undermine the prevailing culture. Therefore, the strategy has to be a SCENE for being transmitted in a stable manner through observational learning.

As a reference point, we have started with the conventional assumption that one can identify a strategy without uncertainty, and checked the best-response relations within the set of $M_1$ pure strategies. Our finding is that a symmetric NE is possible if one uses one of the following three strategies: AllD, $GT_1$ and WSLS (figure 1). Only the last one is efficient. Although we have restricted ourselves to pure strategies, we can discuss the idea behind it as follows: Let us consider a monomorphic population playing a mixed strategy $\mathbf{q} = [q_{CC}, q_{CD}, q_{DC}, q_{DD}]$, where each element means the probability to cooperate in a given circumstance. Such a mixed strategy can be represented as a point inside a four-dimensional unit hypercube. The observer seeks the best response to it, say, $\mathbf{p} = [p_{CC}, p_{CD}, p_{DC}, p_{DD}]$. Suppose that $\mathbf{p}$ also turns out to be a mixed strategy, say, containing $\mathbf{d}_k$ and $\mathbf{d}_l$ with $k \neq l$. According

to the Bishop–Cannings theorem [22], it implies that

$$\Pi(\mathbf{d}_k, \mathbf{q}, \epsilon) = \Pi(\mathbf{d}_l, \mathbf{q}, \epsilon), \tag{3.1}$$

and this equality imposes a set of constraints on $\mathbf{q}$, rendering the dimensionality of the solution manifold lower than four. Therefore, to almost all $\mathbf{q}$ in the four-dimensional hypercube, only one pure strategy will be found as the best response. In appendix C, we provide an explicit proof for this argument in case of reactive strategies.

Even if our theoretical framework of Bayesian best-response dynamics is an idealization, we believe that it captures certain aspects of animal behaviour. For example, although the best-response dynamics *per se* shows poor performance in explaining learning behaviour because of its deterministic character [23], its modified versions can provide reasonable description for experimental results [24,25]. In addition, some studies show that Bayesian updating yields consistent results with observed behaviour of animals, including mammals, birds, a fish and an insect, in the foraging and reproduction activities [26]. These studies support the Bayesian brain hypothesis, which argues that the brain has to successfully simulate the external world in which Bayes' theorem holds [27]. We also point out that the posterior can be calculated correctly even if the observer has short-term memory as implied by the $M_1$ assumption: As long as input observations are exchangeable with each other, Bayesian updating can be done in a sequential manner (i.e. by modifying the prior little by little every time a new observation arrives), and it is mathematically equivalent to a batch update that uses all the observations at once.

To conclude, if we take observational learning into consideration, our result suggests that WSLS can be a SCENE to a Bayesian observer, whereas AllD cannot under observational uncertainty. That is, if the number of observations is too small to see how to behave after error, the uncertainty provides a way to escape from full defection, whereas WSLS can still maintain cooperation: The point is that AllD is not easy to learn by observing defectors because it is difficult to tell what they would choose if someone actually cooperated. WSLS is also difficult to learn, but the uncertainty works in an asymmetric way because one can expect more from mutual cooperation than from full defection by the very definition of the PD game.

Data accessibility. Source codes used in this work are available from the Dryad Digital Repository: https://doi.org/10.5061/dryad.n02v6wwwz [28].

Authors' contributions. J.-K.C. conceived of the study. S.K.B. designed the study and wrote the paper. M.K. analysed the model. All authors gave final approval for publication and agree to be held accountable for the work performed therein.

Competing interests. We declare we have no competing interests.

Funding. M.K. was supported by Basic Science Research Program through the National Research Foundation of Korea (NRF) funded by the Ministry of Education (NRF-2020R1A6A3A13075972). S.K.B. was supported by Basic Science Research Program through the National Research Foundation of Korea (NRF) funded by the Ministry of Education (NRF-2020R1I1A2071670).

# Appendix A. Stationary distribution

Let us consider two players, Alice and Bob, playing the PD game repeatedly. As written in equation (2.2), Alice's effective behaviour in the noisy environment is described by a mixed strategy $\mathbf{s}_A^\epsilon = (q_{CC}, q_{CD}, q_{DC}, q_{DD})$, where $q_{XY} \in \{\epsilon, 1-\epsilon\}$ denotes Alice's probability of cooperation when she and Bob did $X$ and $Y$, respectively, in the previous round. In the same manner, another mixed strategy $\mathbf{s}_B^\epsilon = (r_{CC}, r_{CD}, r_{DC}, r_{DD})$ applies to Bob's effective behaviour (equation (2.3)), where $r_{XY} \in \{\epsilon, 1-\epsilon\}$ denotes Bob's probability of cooperation when he and Alice did $X$ and $Y$, respectively, in the previous round. Let $v_{XY}^{(t)}$ be the probability to see Alice and Bob choosing $X$ and $Y$, respectively, in round $t$. The condition $v_{CC}^{(t)} + v_{CD}^{(t)} + v_{DC}^{(t)} + v_{DD}^{(t)} = 1$ is satisfied all the time. The probability distribution $\mathbf{v}^{(t)} \equiv (v_{CC}^{(t)}, v_{CD}^{(t)}, v_{DC}^{(t)}, v_{DD}^{(t)})$ evolves as $\mathbf{v}^{(t+1)} = W\mathbf{v}^{(t)}$ with

$$W = \begin{bmatrix} q_{CC}r_{CC} & q_{CD}r_{DC} & q_{DC}r_{CD} & q_{DD}r_{DD} \\ q_{CC}\bar{r}_{CC} & q_{CD}\bar{r}_{DC} & q_{DC}\bar{r}_{CD} & q_{DD}\bar{r}_{DD} \\ \bar{q}_{CC}r_{CC} & \bar{q}_{CD}r_{DC} & \bar{q}_{DC}r_{CD} & \bar{q}_{DD}r_{DD} \\ \bar{q}_{CC}\bar{r}_{CC} & \bar{q}_{CD}\bar{r}_{DC} & \bar{q}_{DC}\bar{r}_{CD} & \bar{q}_{DD}\bar{r}_{DD} \end{bmatrix}, \tag{A 1}$$

where $\bar{q}_{XY} \equiv 1 - q_{XY}$ and $\bar{r}_{XY} \equiv 1 - r_{XY}$. Note that it is a positive stochastic matrix for $\epsilon > 0$. According to the Perron–Frobenius theorem, it has a unique largest eigenvalue 1, and the corresponding eigenvector can be chosen to have positive entries. Thus, by solving $W\mathbf{v} = \mathbf{v}$, we can obtain the stationary distribution $\mathbf{v} = (v_{CC}, v_{CD}, v_{DC}, v_{DD})$. Each element $v_{XY}$ can be interpreted as the long-time average frequency of $XY$, and it can readily be expanded as a Taylor series in terms of $\epsilon$. To determine the best response to $\mathbf{d}_\beta$ as shown in table 1, we calculate the long-term average payoff of $\mathbf{d}_\alpha$ against it for every $\alpha \in \{0, \ldots, 15\}$ (equation (2.5)) and compare the Taylor-expanded expressions order by order. As for table 2, we set $\alpha = \beta$ and retain only the leading order terms in the Taylor series for $\mathbf{v}$.

# Appendix B. Bayesian inference

To illustrate the inference procedure, let us assume that $\mathbf{v} \approx (0, 0, 0, 1)$ is given to Alice. She has a set of candidate strategies $\Lambda \equiv \{\mathbf{d}_0, \mathbf{d}_6, \mathbf{d}_8\}$ for the resident strategy $\mathbf{q}$. Alice assigns equal prior probability to each of these candidate strategies. In a certain round $t$, she observes interaction between Eve and Frank both of whom use $\mathbf{q}$. Let $E_t$ and $F_t$ denote Eve's and Frank's moves, respectively, in round $t$. If Alice sees Eve cooperate (i.e. $E_t = C$) after $S_{t-1} \equiv (E_{t-1}, F_{t-1}) = (C, C)$, she may use this additional information in a Bayesian way to calculate the posterior probability of $\mathbf{q} = \mathbf{d}_0$ as follows:

$$P(\mathbf{q} = \mathbf{d}_0 | E_t, S_{t-1}) = \frac{P(E_t | S_{t-1}, \mathbf{d}_0)P(S_{t-1} | \mathbf{d}_0)P(\mathbf{d}_0)}{\sum_{\mathbf{d}_i \in \Lambda} P(E_t | S_{t-1}, \mathbf{d}_i)P(S_{t-1} | \mathbf{d}_i)P(\mathbf{d}_i)} \tag{B 1}$$

$$= \frac{\epsilon \cdot \epsilon^2 \cdot (1/3)}{\epsilon \cdot \epsilon^2 \cdot (1/3) + \epsilon \cdot (2\epsilon - 5\epsilon^2 + 4\epsilon^3) \cdot (1/3) + \epsilon \cdot \epsilon/2 \cdot (1/3)}, \tag{B 2}$$

where $P(E_t | S_{t-1}, \mathbf{d}_i)$ is directly obtained from $\mathbf{d}_i$, and $P(S_{t-1} | \mathbf{d}_i)$ is taken from the stationary probability distribution $\mathbf{v}$. This posterior probability is used as prior probability for the next observation. If $\mathbf{q}$ is actually $\mathbf{d}_6$, the average number of times to observe $E_t = C$ after $S_{t-1} = (C, C)$ will be

$$MP(E_t, S_{t-1} | \mathbf{q} = \mathbf{d}_6) = MP(E_t | S_{t-1}, \mathbf{d}_6)P(S_{t-1} | \mathbf{d}_6). \tag{B 3}$$

In this way, Alice obtains the final posterior probability of $\mathbf{q} = \mathbf{d}_0$ after observing interaction between $M$ pairs of players, when their actual strategy is $\mathbf{d}_6$. If $\epsilon$ is fixed as a small positive value, this inference procedure approaches the correct answer as $M \to \infty$. The effect of observational uncertainty manifests

itself when $M\epsilon \ll 1$. For example, we may choose $M \approx \epsilon^{-1/2}$ as a representative value for $1 \ll M \ll \epsilon^{-1}$ and check various values of $\epsilon$ from $10^{-2}$ to $10^{-6}$. Then, the above calculation confirms that the posterior probabilities should remain identical to the prior ones due to the lack of observation.

## Appendix C. Best-response relations among reactive strategies

Let us consider two reactive strategies $\mathbf{p} = [p_c, p_D, p_c, p_D]$ and $\mathbf{q} = [q_c, q_D, q_c, q_D]$. The long-term average payoff of $\mathbf{p}$ against $\mathbf{q}$ is

$$\Pi = \frac{(p_D q_c - p_D q_D + q_D) - c(p_D + p_c q_D - p_D q_D)}{1 - (p_c - p_D)(q_c - q_D)} \quad (C1)$$

in the limit of $\epsilon \to 0$. After some algebra, we find the following: First, if $q_C - q_D > c$, both $\partial\Pi/\partial p_C$ and $\partial\Pi/\partial p_D$ are positive, so the best response is given by $p_c = p_D = 1$. Or, if $q_C - q_D < c$, both $\partial\Pi/\partial p_C$ and $\partial\Pi/\partial p_D$ are negative, so the best response is given by $p_c = p_D = 0$. Note that we have neglected the measure-zero line defined by $q_C - q_D = c$, on which the best response is not uniquely determined.

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
