## [Peer Review File · Proceedings of the Royal Society B: Biological Sciences]

Review History

RSPB-2021-0047.R0 (Original submission)

Review form: Reviewer 1

Recommendation

Accept with minor revision (please list in comments)

Scientific importance: Is the manuscript an original and important contribution to its field?

Good

General interest: Is the paper of sufficient general interest?

Good

Quality of the paper: Is the overall quality of the paper suitable?

Good

Is the length of the paper justified?

Yes

Should the paper be seen by a specialist statistical reviewer?

No

Do you have any concerns about statistical analyses in this paper? If so, please specify them explicitly in your report.

No

It is a condition of publication that authors make their supporting data, code and materials available - either as supplementary material or hosted in an external repository. Please rate, if applicable, the supporting data on the following criteria.

Is it accessible?

N/A

Is it clear?

N/A

Is it adequate?

N/A

Do you have any ethical concerns with this paper?

No

Comments to the Author

In this manuscript, the authors investigated the iterated PD game in terms of best-response relations and the dynamics by adding the mechanism of observational learning. They assume that each player cannot know their opponents' strategies but has memory-one stochastic strategies in the iterated prisoner's dilemma games. They find that players can escape from full defection into a cooperative equilibrium supported by Win-Stay Lose-Shift in a self-confirming manner. I have found it clear and comprehensive. I think that the results are convincing and give significant values to the audience. Thus I support the publication of the work in the journal. However, I still have some following comments on this work.

- (1) It is not very clear to me from the model description how players calculate the best response strategy. I suggest the authors clarify it.
- (2) I do not think the authors have provide all the cases of strategies which can evolve to WSLS.
- (3) In the model, players use a very smart mechanism to update their strategies. And I am concern that if all of participants are so smart. I mean if each player wants to calculate a best response strategy, what is the final state of the population?

Review form: Reviewer 2

Recommendation

Major revision is needed (please make suggestions in comments)

Scientific importance: Is the manuscript an original and important contribution to its field?

Good

General interest: Is the paper of sufficient general interest?

Good

Quality of the paper: Is the overall quality of the paper suitable?

Good

Is the length of the paper justified?

Yes

Should the paper be seen by a specialist statistical reviewer?

No

Do you have any concerns about statistical analyses in this paper? If so, please specify them explicitly in your report.

No

It is a condition of publication that authors make their supporting data, code and materials available - either as supplementary material or hosted in an external repository. Please rate, if applicable, the supporting data on the following criteria.

Is it accessible?

N/A

Is it clear?

N/A

Is it adequate?

N/A

Do you have any ethical concerns with this paper?

No

Comments to the Author

A key element in the specification of an evolutionary game dynamic is the rule that describes what strategy a focal individual will adopt, leading to a potentially new composition of the population. In a biological context, one often assumes that the focal individual is an offspring that inherits its strategy. However, if the dynamic describes cultural evolution, the focal individual may select or revise his strategy by using information about the current state of the population. The paper under review analyses, in the context of a repeated prisoner's dilemma game, the impact of the amount of information that is available to the focal individual. First, for comparison, the authors consider the standard case where the individual has perfect information and chooses a corresponding best response. Then they turn to the case where the focal individual is a Bayesian learner that can only make a finite number of observations on the behavior in the current population and bases the strategy choice on the posterior. The main insight is that, in contrast to the case of full information, with little information, the strategy "Always Defect" need not be a self-confirming equilibrium and so the population can move to a cooperative state where the strategy "Win-Stay-Lose-Shift" is used.

I think the manuscript deals with an important topic, but in my opinion the Bayesian analysis should be extended. The authors consider only two cases. In the first case, the Bayesian learner has sufficiently many observations so that there is practically no difference to having complete information. In the other case, there are so few observations that the Bayesian learner bases his choice just on his prior. I acknowledge that the support of the prior depends on the observations. Still, I think that a Bayesian would be reluctant to rely on his prior without updating. One can argue that the focal individual must come to a decision and if he has no information to update his prior, he would be forced to use only the prior. What is more problematic is that the main conclusion of the paper depends on the choice of the prior, which is here taken to be a uniform distribution. This is a convenient choice, but somewhat arbitrary.

In the abstract, the authors write the observer has to "adjust" his strategy. This suggests that he has already been using a strategy and it seems more plausible that without the possibility of updating the prior he would continue using that strategy. Is the main result still true under this modification?

To clarify the role of the prior it would be good to extend the discussion around equations (2.6) to (2.9) to arbitrary priors. For example, for selected values of c , a figure could be included that shows for each prior distribution on the three candidates d_0 , d_6 , d_8 the resulting best response (Every such distribution can be represented by a point in a triangle.) Similarly for priors on the candidates d_9 , d_{14} , d_{15} . The extended discussion should show the robustness or otherwise of the main conclusion to the choice of the prior.

Minor comments:

Sandholm [International Journal of Game Theory 30 (2001) 107-116] and Kreindler and Young [Games and Economic Behavior 80 (2013) 39-67] consider game dynamics where the revising agent can use only a sample of a given size and they study how the dynamics depend on this size. It is perhaps useful to relate the present results to their approach.

The figures and the tables are based on many lengthy calculations. It might be helpful for a reader to include some of these calculations in an electronic appendix.

Decision letter (RSPB-2021-0047.R0)

01-Mar-2021

Dear Dr Baek:

I am writing to inform you that your manuscript RSPB-2021-0047 entitled "Win-Stay-Lose-Shift as a self-confirming equilibrium in the iterated prisoner's dilemma" has, in its current form, been rejected for publication in Proceedings B.

This action has been taken on the advice of referees, who have recommended that substantial revisions are necessary. With this in mind we would be happy to consider a resubmission, provided the comments of the referees are fully addressed. However please note that this is not a provisional acceptance.

To upload a resubmitted manuscript, log into <http://mc.manuscriptcentral.com/prsb> and enter your Author Centre, where you will find your manuscript title listed under "Manuscripts with

Decisions." Under "Actions," click on "Create a Resubmission." Please be sure to indicate in your cover letter that it is a resubmission, and supply the previous reference number.

Sincerely,
Dr Robert Barton
mailto: proceedingsb@royalsociety.org

Associate Editor
Board Member: 1
Comments to Author:

The manuscript under consideration was reviewed by two experts and myself. We all found the study interesting, with potentially important findings about how learning influences strategies in iterated games. This is an understudied topic and could be a way to make game-theoretical models more applicable to real world behavior, particularly human behavior. The specific finding that imperfect learning can favor Win-Stay-Lose-Shift over All Defect is particularly exciting. However, Reviewer 2 has raised a serious concern about the robustness of the current model to changes in the prior distribution for the Bayesian updating. This is an important concern as it would be unfortunate if the broader conclusion that learning can favor more cooperation is only relevant to a particular choices of model features. This issue must be addressed before the manuscript can be published. Reviewer 2 makes suggestions for how to do this and which will require additional runs of the model and new data collection. In addition, Reviewer 1 requests further information about how the individuals calculate the best strategies and how this ability relates to the way the model plays out. I also see a need to explain a related issue: if animals are able to calculate the best response strategy in a fairly sophisticated way, how does that fit with some of the other assumptions of the model (e.g. that they respond based solely on the previous trial)? Does the model describe a realistic collection of behaviors? In general, more needs to be done to 1) demonstrate that the findings are broadly applicable to a variety of situations, and 2) help the reader see that this is true by providing more details about how the model works and how this captures realistic situations.

Reviewer(s)' Comments to Author:

Referee: 1

Comments to the Author(s)

In this manuscript, the authors investigated the iterated PD game in terms of best-response relations and the dynamics by adding the mechanism of observational learning. They assume that each player cannot know their opponents' strategies but has memory-one stochastic strategies in the iterated prisoner's dilemma games. They find that players can escape from full defection into a cooperative equilibrium supported by Win-Stay Lose-Shift in a self-confirming manner. I have found it clear and comprehensive. I think that the results are convincing and give significant values to the audience. Thus I support the publication of the work in the journal. However, I still have some following comments on this work.

- (1) It is not very clear to me from the model description how players calculate the best response strategy. I suggest the authors clarify it.
- (2) I do not think the authors have provide all the cases of strategies which can evolve to WSLS.
- (3) In the model, players use a very smart mechanism to update their strategies. And I am concern that if all of participants are so smart. I mean if each player wants to calculate a best response strategy, what is the final state of the population?

Referee: 2

Comments to the Author(s)

A key element in the specification of an evolutionary game dynamic is the rule that describes what strategy a focal individual will adopt, leading to a potentially new composition of the population. In a biological context, one often assumes that the focal individual is an offspring that inherits its strategy. However, if the dynamic describes cultural evolution, the focal individual may select or revise his strategy by using information about the current state of the population.

The paper under review analyses, in the context of a repeated prisoner's dilemma game, the impact of the amount of information that is available to the focal individual. First, for comparison, the authors consider the standard case where the individual has perfect information and chooses a corresponding best response. Then they turn to the case where the focal individual is a Bayesian learner that can only make a finite number of observations on the behavior in the current population and bases the strategy choice on the posterior. The main insight is that, in contrast to the case of full information, with little information, the strategy "Always Defect" need not be a self-confirming equilibrium and so the population can move to a cooperative state where the strategy "Win-Stay-Lose-Shift" is used.

I think the manuscript deals with an important topic, but in my opinion the Bayesian analysis should be extended. The authors consider only two cases. In the first case, the Bayesian learner has sufficiently many observations so that there is practically no difference to having complete information. In the other case, there are so few observations that the Bayesian learner bases his choice just on his prior. I acknowledge that the support of the prior depends on the observations. Still, I think that a Bayesian would be reluctant to rely on his prior without updating. One can argue that the focal individual must come to a decision and if he has no information to update his prior, he would be forced to use only the prior. What is more problematic is that the main conclusion of the paper depends on the choice of the prior, which is here taken to be a uniform distribution. This is a convenient choice, but somewhat arbitrary.

In the abstract, the authors write the observer has to "adjust" his strategy. This suggests that he has already been using a strategy and it seems more plausible that without the possibility of updating the prior he would continue using that strategy. Is the main result still true under this modification?

To clarify the role of the prior it would be good to extend the discussion around equations (2.6) to (2.9) to arbitrary priors. For example, for selected values of c , a figure could be included that shows for each prior distribution on the three candidates d_0, d_6, d_8 the resulting best response (Every such distribution can be represented by a point in a triangle.) Similarly for priors on the candidates d_9, d_{14}, d_{15} . The extended discussion should show the robustness or otherwise of the main conclusion to the choice of the prior.

Minor comments:

Sandholm [International Journal of Game Theory 30 (2001) 107-116] and Kreindler and Young [Games and Economic Behavior 80 (2013) 39-67] consider game dynamics where the revising agent can use only a sample of a given size and they study how the dynamics depend on this size. It is perhaps useful to relate the present results to their approach.

The figures and the tables are based on many lengthy calculations. It might be helpful for a reader to include some of these calculations in an electronic appendix.

Author's Response to Decision Letter for (RSPB-2021-0047.R0)

See Appendix A.

RSPB-2021-1021.R0

Review form: Reviewer 2

Recommendation

Accept with minor revision (please list in comments)

Scientific importance: Is the manuscript an original and important contribution to its field?

Good

General interest: Is the paper of sufficient general interest?

Good

Quality of the paper: Is the overall quality of the paper suitable?

Good

Is the length of the paper justified?

Yes

Should the paper be seen by a specialist statistical reviewer?

No

Do you have any concerns about statistical analyses in this paper? If so, please specify them explicitly in your report.

No

It is a condition of publication that authors make their supporting data, code and materials available - either as supplementary material or hosted in an external repository. Please rate, if applicable, the supporting data on the following criteria.

Is it accessible?

N/A

Is it clear?

N/A

Is it adequate?

N/A

Do you have any ethical concerns with this paper?

No

Comments to the Author

I think the new material in the revision is very helpful. In particular, it is good to see that the main conclusions are robust to changes in the prior distribution.

I have only two minor remarks.

In the added discussion (line 130) it is somewhat misleading to speak of "an observer of an AllD population". AllD is a particular strategy, d_0 , and the observer does not know whether this strategy is being used or d_6 or d_8 . Perhaps it would be better to avoid the abbreviation and to speak of an observer that sees nearly only defection. A similar comment applies to AllC and AllD in lines 138 and 139 as well as in lines 3 and 6 of the text explaining Figure 3.

At the beginning of the paper there occurs ``Proceedings A" and ``rspa ...". I believe this is a mistake.

Decision letter (RSPB-2021-1021.R0)

01-Jun-2021

Dear Dr Baek

I am pleased to inform you that your manuscript RSPB-2021-1021 entitled "Win-Stay-Lose-Shift as a self-confirming equilibrium in the iterated prisoner's dilemma" has been accepted for publication in Proceedings B.

The referee(s) have recommended publication, but also suggest some minor revisions to your manuscript. Therefore, I invite you to respond to the referee(s)' comments and revise your manuscript. Because the schedule for publication is very tight, it is a condition of publication that you submit the revised version of your manuscript within 7 days. If you do not think you will be able to meet this date please let us know.

Online supplementary material will also carry the title and description provided during submission, so please ensure these are accurate and informative. Note that the Royal Society will not edit or typeset supplementary material and it will be hosted as provided. Please ensure that

the supplementary material includes the paper details (authors, title, journal name, article DOI). Your article DOI will be 10.1098/rspb.[paper ID in form xxxx.xxxx e.g. 10.1098/rspb.2016.0049].

Sincerely,

Dr Robert Barton

Associate Editor

Board Member

Comments to Author:

Thank you for your careful revision and response to reviewers. Note that Reviewer 1 makes some wording suggestions and points to a mistake--please make those corrections before submitting the final version. In addition, I think starting the manuscript with a reference to nature and nurture might be off-putting to some readers (it feels like a dated way to describe causes of variation). I suggest deleting the initial clause and just saying "Evolutionary game theorists often assume that behavioral traits...".

Reviewer(s)' Comments to Author:

Referee: 2

Comments to the Author(s).

I think the new material in the revision is very helpful. In particular, it is good to see that the main conclusions are robust to changes in the prior distribution.

I have only two minor remarks.

In the added discussion (line 130) it is somewhat misleading to speak of "an observer of an AllD population". AllD is a particular strategy, d_0 , and the observer does not know whether this strategy is being used or d_6 or d_8 . Perhaps it would be better to avoid the abbreviation and to speak of an observer that sees nearly only defection. A similar comment applies to AllC and AllD in lines 138 and 139 as well as in lines 3 and 6 of the text explaining Figure 3.

At the beginning of the paper there occurs "Proceedings A" and "rspa ...". I believe this is a mistake.

Author's Response to Decision Letter for (RSPB-2021-1021.R0)

See Appendix B.

Decision letter (RSPB-2021-1021.R1)

04-Jun-2021

Dear Dr Baek

I am pleased to inform you that your manuscript entitled "Win-Stay-Lose-Shift as a self-confirming equilibrium in the iterated prisoner's dilemma" has been accepted for publication in Proceedings B.

Data Accessibility section

Open Access

Paper charges

Sincerely,

Appendix A

Pukyong National University
Department of Physics
Busan 48513, Korea
SEUNG KI BAEK
Email: seungki@pknu.ac.kr
Tel: +82 51 629 5576
Fax: +82 51 629 5549

April 20, 2021

Dear Editor,

We are pleased to see that both the reviewers found our work “convincing” and “important.” We have tried our best to answer their questions and comments as detailed below. We hope that our revised manuscript is now suitable for publication in *Proceedings of the Royal Society B*.

Yours Sincerely,

Seung Ki Baek
on behalf of the authors

To Associate Editor

Editor: The manuscript under consideration was reviewed by two experts and myself. We all found the study interesting, with potentially import [sic] findings about how learning influences strategies in iterated games. This is an understudied topic and could be a way to make game-theoretical models more applicable to real world behavior, particularly human behavior. The specific finding that imperfect learning can favor Win-Stay-Lose-Shift over All Defect is particularly exciting.

Answer: We are very grateful for your thoughtful summary of the reviews.

Editor: However, Reviewer 2 has raised a serious concern about the robustness of the current model to changes in the prior distribution for the Bayesian updating. This is an important concern as it would be unfortunate if the broader conclusion that learning can favor more cooperation is only relevant to a particular choices of model features. This issue must be addressed before the manuscript can be published. Reviewer 2 makes suggestions for how to do this and which will require additional runs of the model and new data collection.

Answer: Yes, this is indeed an important concern. As you will see below, we have followed Reviewer 2's suggestions to answer this comment (List of changes **#2, #3, #4, and #5**).

Editor: In addition, Reviewer 1 requests further information about how the individuals calculate the best strategies and how this ability relates to the way the model plays out. I also see a need to explain a related issue: if animals are able to calculate the best response strategy in a fairly sophisticated way, how does that fit with some of the other assumptions of the model (e.g. that they respond based solely on the previous trial)? Does the model describe a realistic collection of behaviors?

Answer: First of all, Reviewer 1's request will be answered below (List of changes **#2 and #3**).

As for your question, our theoretical framework is certainly an idealization, but we believe that it captures certain aspects of reality. For example, in Van Huyck et al. (1997), human learning behaviour is well fitted to an approximate version of the best-response dynamics. When it comes to Bayesian updating, according to a review article which we have added to References as [25], all of 11 empirical studies except one show consistent results with Bayesian models. The Bayesian brain hypothesis, whose history goes back to the 1860s when Hermann von Helmholtz developed experimental psychology, actually argues that the brain has to successfully simulate the external world in which Bayes' theorem holds. We would also like to point out that the Bayesian idea does not contradict with the restriction to M_1 strategies because one can sequentially update the prior little by little by referring to the latest observation, which is consistent with the M_1 assumption, and the result is mathematically equivalent to that of a batch update. We have added more discussion on your question to Summary and Discussion (List of changes #1).

Editor: In general, more needs to be done to 1) demonstrate that the findings are broadly applicable to a variety of situations, and 2) help the reader see that this is true by providing more details about how the model works and how this captures realistic situations.

Answer: We agree. The answers will be given in full detail below.

To Reviewer #1

Reviewer: In this manuscript, the authors investigated the iterated PD game in terms of best-response relations and the dynamics by adding the mechanism of observational learning. They assume that each player cannot know their opponents strategies but has memory-one stochastic strategies in the iterated prisoners dilemma games. They find that players can escape from full defection into a cooperative equilibrium supported by Win-Stay Lose-Shift in a self-confirming [sic] manner. I have found it clear and comprehensive. I think that the results are convincing and give significant values to the audience. Thus I support the publication of the work in the journal.

Answer: We are grateful for your careful reading and insightful questions, which we answer below.

Reviewer: However, I still have some following comments on this work.

- (1) It is not very clear to me from the model description how players calculate the best response strategy. I suggest the authors clarify it.
- (2) I do not think the authors have provide all the cases of strategies which can evolve to WSLS.
- (3) In the model, players use a very smart mechanism to update their strategies. And I am concern that if all of participants are so smart. I mean if each player wants to calculate a best response strategy, what is the final state of the population?

Answer: (1) We have newly added an appendix to clarify how we calculate the best response (List of changes #2).

(2) We believe that you are asking the condition for the prior to expect WSLS as the best response. We have explained it at the end of Method and Result in this revised manuscript (List of changes #3).

(3) If everyone calculates and adopts the best response to the existing strategy in the population at a given time step, the population will change their strategy all at once at the next time step. According to Fig. 2, if $c < 2/9$, the population will eventually end up with WSLS.

To Reviewer #2

Reviewer: A key element in the specification of an evolutionary game dynamic is the rule that describes what strategy a focal individual will adopt, leading to a potentially new composition of the population. In a biological context, one often assumes that the focal individual is an offspring that inherits its strategy. However, if the dynamic describes cultural evolution, the focal individual may select or revise his strategy by using information about the current state of the population. The paper under review analyses, in the context of a repeated prisoner's dilemma game, the impact of the amount of information that is available to the focal individual. First, for comparison, the authors consider the standard case where the individual has perfect information and chooses a corresponding best response. Then they turn to the case where the focal individual is a Bayesian learner that can only make a finite number of observations on the behavior in the current population and bases the strategy choice on the posterior. The main insight is that, in contrast to the case of full information, with little information, the strategy "Always Defect" need not be a self-confirming equilibrium and so the population can move to a cooperative state where the strategy "Win-Stay-Lose-Shift" is used.

Answer: We are grateful for your careful reading and constructive comments.

Reviewer: I think the manuscript deals with an important topic, but in my opinion the Bayesian analysis should be extended. The authors consider only two cases. In the first case, the Bayesian learner has sufficiently many observations so that there is practically no difference to having complete information. In the other case, there are so few observations that the Bayesian learner bases his choice just on his prior. I acknowledge that the support of the prior depends on the observations. Still, I think that a Bayesian would be reluctant to rely on his prior without updating. One can argue that the focal individual must come to a decision and if he has no information to update his prior, he would be forced to use only the prior. What is more problematic is that the main conclusion of the paper depends on the choice of the prior, which is here taken to be a uniform distribution. This is a convenient choice, but somewhat arbitrary.

Answer: This is certainly an important point in the Bayesian analysis in general. As you pointed out, our assumption is that the focal individual has to make a decision based on the prior if no more information is available from observation, which means that the decision will heavily depend on the prior. As will be shown below, we have conducted a more detailed analysis on the effect of the prior as you suggested.

Reviewer: In the abstract, the authors write the observer has to “adjust” his strategy. This suggests that he has already been using a strategy and it seems more plausible that without the possibility of updating the prior he would continue using that strategy. Is the main result still true under this modification?

Answer: This is a sharp comment. As we explained above, our assumption is not that the focal individual starts with a certain strategy to revise upon the arrival of information but that he or she has to calculate the best response from the prior in the first place. We regret that we were unclear about it by choosing the word ‘adjust’. It has been changed to ‘chooses’ in this revised manuscript (List of changes #4).

Reviewer: To clarify the role of the prior it would be good to extend the discussion around equations (2.6) to (2.9) to arbitrary priors. For example, for selected values of c , a figure could be included that shows for each prior distribution on the three candidates d_0 , d_6 , d_8 the resulting best response (Every such distribution can be represented by a point in a triangle.) Similarly for priors on the candidates d_9 , d_{14} , d_{15} . The extended discussion should show the robustness or otherwise of the main conclusion to the choice of the prior.

Answer: This is a great idea. We have carried out this extended calculation and added it to Method and Result (List of change #3).

Reviewer: Sandholm [International Journal of Game Theory 30 (2001) 107-116] and Kreindler and Young [Games and Economic Behavior 80 (2013) 39-67] consider game dynamics where the revising agent can use only a sample of a given size and they study how the dynamics depend on this size. It is perhaps useful to relate the present results to their approach.

Answer: The suggested references shows a degree of similarity to our manuscript in that they consider strategy revision based on imperfect observations. Still, an important difference is that strategies are indistinguishable from actions in their coordination game, so that the players are actually able to observe others' strategies almost directly. In their works, the source of imperfection rather originates from the inhomogeneity of the population, whereas we have been considering a monomorphic population (List of changes **#5**).

Reviewer: The figures and the tables are based on many lengthy calculations. It might be helpful for a reader to include some of these calculations in an electronic appendix.

Answer: This is another great idea. We have newly added an appendix to explain to readers how we calculate the best response (List of changes **#2**).

List of changes

#1 In Summary and Discussion, we have added the following paragraph:

Even if our theoretical framework of Bayesian best-response dynamics is an idealization, we believe that it captures certain aspects of animal behaviour. For example, although the best-response dynamics *per se* shows poor performance in explaining learning behaviour because of its deterministic character [22], its modified versions can provide reasonable description for experimental results [23,24]. In addition, some studies show that Bayesian updating yields consistent results with observed behaviour of animals, including mammals, birds, a fish and an insect, in the foraging and reproduction activities [25]. These studies support the Bayesian brain hypothesis, which argues that the brain has to successfully simulate the external world in which Bayes' theorem holds [26]. We also point out that the posterior can be calculated correctly even if the observer has short-term memory as implied by the M_1 assumption: As long as input observations are exchangeable with each other, Bayesian updating can be done in a sequential manner, i.e., by modifying the prior little by little every time a new observation arrives, and it is mathematically equivalent to a batch update that uses all the observations at once.

#2 We have newly added Appendix A to explain more details of the best-response calculation.

#3 At the end of Method and Result, this revised manuscript presents conditions of the prior for WSLs to give the highest expected payoff when observed behaviour is close to full defection or full cooperation. We have also newly added Fig. 3 to visualize the conditions.

#4 In Abstract, we replace 'adjust' by 'chooses' as follows:

Based on the observation, the observer has to infer the resident strategy in a Bayesian way and *chooses* his or her own strategy accordingly.

#5 In Introduction, we have added the following sentences:

Dynamics of learning based on a limited set of information has been investigated in the context of the coordination game [12,13], in which the opponent's observed decision is assumed to be his or her strategy. However, the subtlety of cultural transmission manifests itself clearly when a strategy is regarded as a decision rule, hidden from the observer, rather than the decision itself.

#6 We have revised some expressions to improve readability.

Appendix B

Pukyong National University
Department of Physics
Busan 48513, Korea
SEUNG KI BAEK
Email: seungki@pknu.ac.kr
Tel: +82 51 629 5576
Fax: +82 51 629 5549

June 4, 2021

Dear Editor,

We are pleased with the acceptance of our paper. The additional comments are again very helpful, and we have revised the manuscript accordingly.

Yours Sincerely,

Seung Ki Baek
on behalf of the authors

To Associate Editor

Editor: Thank you for your careful revision and response to reviewers. Note that Reviewer 1 makes some wording suggestions and points to a mistake—please make those corrections before submitting the final version. In addition, I think starting the manuscript with a reference to nature and nurture might be off-putting to some readers (it feels like a dated way to describe causes of variation). I suggest deleting the initial clause and just saying “Evolutionary game theorists often assume that behavioral traits...”

Answer: We are grateful for your helpful comment. We have revised the opening sentence as you suggested (List of changes #1).

To Reviewer #2

Reviewer: I think the new material in the revision is very helpful. In particular, it is good to see that the main conclusions are robust to changes in the prior distribution. I have only two minor remarks.

Answer: We are so thankful for your review report, which has greatly improved this manuscript. We answer your comments below.

Reviewer: In the added discussion (line 130) it is somewhat misleading to speak of “an observer of an AllD population”. AllD is a particular strategy, d_0 , and the observer does not know whether this strategy is being used or d_6 or d_8 . Perhaps it would be better to avoid the abbreviation and to speak of an observer that sees nearly only defection. A similar comment applies to AllC and AllD in lines 138 and 139 as well as in lines 3 and 6 of the text explaining Figure 3.

Answer: This is a great comment, and we definitely agree on it. The expressions have been revised according to your comment (List of changes #2).

Reviewer: At the beginning of the paper there occurs “Proceedings A” and “rspa ...”. I believe this is a mistake.

Answer: Unfortunately, we have been unable to find a LaTeX template for Proceedings B. We hope that the manuscript will be properly reformatted in the production stage.

List of changes

#1 The sentence contrasting nature and nurture has been deleted, so Introduction begins as follows:

“Evolutionary game theorists often assume that behavioural traits can be genetically transmitted across generations.”

#2 In the added discussion, the remarks on particular strategies have been replaced by the following expressions:

“For an observer who almost always sees defection from the population, ...”

“ Similarly, we can check what an observer would conclude after observing nearly cooperation only, although it is of less importance compared with the above case of a defecting population.”

The caption of Fig. 3 has also been revised as follows:

“When the observer sees nearly defection only, ...”

“If the observer sees cooperation almost all the time, ...”

#3 We have revised the Data Accessibility section to refer to the Dryad Digital Repository.

#4 The second author’s affiliation has been changed.